# High Hydrostatic Pressure in the Modulation of Enzymatic and Organocatalysis and Life under Pressure: A Review

**DOI:** 10.3390/molecules28104172

**Published:** 2023-05-18

**Authors:** Hana Scepankova, Diogo Galante, Edelman Espinoza-Suaréz, Carlos A. Pinto, Letícia M. Estevinho, Jorge Saraiva

**Affiliations:** 1LAQV-REQUIMTE, Department of Chemistry, University of Aveiro, 3810-193 Aveiro, Portugal; 2CIMO, Mountain Research Center Polytechnic Institute of Bragança, Campus Santa Apolónia, 5301-855 Bragança, Portugal

**Keywords:** enzyme, activity, high hydrostatic pressure, equilibrium, substrate

## Abstract

The interest in high hydrostatic pressure (HHP) is mostly focused on the inactivation of deleterious enzymes, considering the quality-related issues associated with enzymes in foods. However, more recently, HHP has been increasingly studied for several biotechnological applications, including the possibility of carrying out enzyme-catalyzed reactions under high pressure. This review aims to comprehensively present and discuss the effects of HHP on the kinetic catalytic action of enzymes and the equilibrium of the reaction when enzymatic reactions take place under pressure. Each enzyme can respond differently to high pressure, mainly depending on the pressure range and temperature applied. In some cases, the enzymatic reaction remains significantly active at high pressure and temperature, while at ambient pressure it is already inactivated or possesses minor activity. Furthermore, the effect of temperature and pressure on the enzymatic activity indicated a faster decrease in activity when elevated pressure is applied. For most cases, the product concentration at equilibrium under pressure increased; however, in some cases, hydrolysis was preferred over synthesis when pressure increased. The compiled evidence of the effect of high pressure on enzymatic activity indicates that pressure is an effective reaction parameter and that its application for enzyme catalysis is promising.

## 1. Introduction

Enzymatic technologies have grown in the past decades as a greener alternative towards the development of more environmentally conscious processes, allowing reactions to occur in milder conditions with increased productivity and selectivity. Nowadays, enzymes are part of many industrial processes, and the demand for more stable, highly active, and specific enzymes is increasing. By the year 2022, industrial enzymes had already reached a global market value of US $12.1 bn, and it is expected to reach US $16.9 bn in 2027, at a compound annual growth rate (CAGR) of 6.8% [1]. Enzymes used in the food and beverage sector comprise the largest segment of the industrial enzymes industry, with a prognosis to grow to US $3.1 bn by 2028, indicating a CAGR of 6.2% [2]. 

The discovery of piezophile organisms, including various thermophiles and hyperthermophiles [3], in extreme conditions of deep-sea environments (up to 120 MPa, 2–100 °C, absence of sunlight, and insufficient supply of organic nutrients) is one of the motivations for the study of enzymes under high pressure [4]. Enzymes from deep-sea thermophiles became useful models for studying the influence of pressure on enzyme activity and stability under extreme physical conditions [4,5,6,7], including high pressure and high temperature [8].

The experimental effect of pressure on enzymatic activity has a long history, beginning with Eyring’s invertase study in 1946 [9]. The first theory about the behavior of enzymes under high pressure was proposed by Laidler (1950) [10], and over 30 years later, the research evolved with Morild (1981) [11], who made an extensive review of the theoretical knowledge known until the present date of the pressure effects on enzymes. A year later, Mozhaev (1994) [12] described the high hydrostatic pressure (HHP) effects on proteins and other biomolecules [13]. 

HHP technology has received significant interest worldwide in the food industry in the last few decades. Indeed, HHP is a novel non-thermal method of food pasteurization with several advantages, including the capacity to inactivate microorganisms while better preserving the nutritional and bioactive components of foods [14]. Many HHP-treated products are already available on the market. Furthermore, HHP has also been shown to be able to beneficially change food texture and influence enzymatic reactions [15]. More recently, this technique has been studied for several biotechnological applications, including the possibility of carrying out enzyme-catalyzed reactions under HHP. HHP can influence (bio)chemical reactions in two different ways: (i) by changing the reaction’s thermodynamical equilibrium [9,16] and (ii) by changing its kinetics [17]. The former may offer the possibility of a higher desired product concentration in the equilibrium state, and the latter may allow the reaction system to reach equilibrium in a shorter time. In this way, more desired products could be obtained in a shorter time when compared to the reaction performed at atmospheric pressure. HHP has also been recently applied to organocatalysis [18]. This type of catalysis uses organic compounds called organocatalysts made up of carbon, hydrogen, nitrogen, oxygen, phosphorus, sulfur, and other nonmetal elements to increase the speed of a chemical reaction. The main advantage of organocatalysis is the possibility of asymmetric synthesis that produces enantiomers in different proportions, which was recently explored by Benjamin List and David MacMillan, who earned the Nobel Prize in Chemistry in 2021 [19,20].

As regards the stability of enzymes, high pressure can be used as a thermal stabilizer to extend the temperature range for a variety of heat-stable enzymes, namely α-amylase [5], thermolysine [17,21,22], alcohol dehydrogenase [21,23], or β-galactosidase [24,25]. 

Previous literature reviews have focused on the description of the reduction of enzymatic activity due to HPP processing related to food quality and preservation [26,27,28], but no review has extensively dealt with the behavior of enzyme activity under HHP pressure. This review aims to compile and describe the effect of HHP on enzymatic activity and other promising catalysts, such as organocatalysts, and to present the main possible advantages of biocatalytic reactions under pressure. Section 1 of this review will discuss the effects of HHP on the thermodynamics and kinetics of an enzyme-catalyzed reaction. Section 2 will give the reader an up-to-date overview of the effect of HHP on the activity of particular enzymes. Section 3 will discuss the potential of performing reactions accelerated by organocatalysts under pressure. Lastly, Section 4 will describe life’s adaptation to HHP environments in the context of enzyme adaptations. 

## 2. Effect of HHP on Enzymatic Systems

The protein structure, state of solvent media, thermodynamics, and kinetics of enzymatic systems can be affected by HHP. In this study, only the thermodynamic and kinetic effects will be discussed because the effect of pressure on protein structure (namely protein denaturation and enzyme pressure inactivation that cause a decrease in enzyme activity) has been extensively reviewed in some papers before [29], and the effect of pressure on solvent media ultimately reflects on the thermodynamics and kinetics of enzyme-catalyzed reactions.

### 2.1. Effect of HHP on Thermodynamics

Chemical transformations imply atom rearrangements, bond breaking, and bond formation that may cause volume changes in the system. The effect of pressure on the thermodynamic equilibrium of the reaction is determined by the whole system reaction volume (∆*V*) [30], defined as the volume change from the final state (volume of product and final solvent media) to the initial state (volume of reactants and initial solvent media) (Equation (1)) [16,31]:(1)$$\Delta V=(V_{product}+V_{final\;solvent\;media})-(V_{reactants}+V_{initial\;solvent\;media})$$

The value for ∆*V* can be calculated using Equation (2) [30]: (2)$${\left(\frac{\partial \ln K}{\partial \;p}\right)}_{T}=-\frac{\Delta \;V}{RT}$$

Furthermore, the pressure effects on an equilibrium constant *K* are determined by ∆*V*, according to Equation (3):(3)$$In\;K=\ln\left(A\right)-\frac{p\;\Delta \;V}{RT}$$
where *R* is the ideal gas constant, *T* is the absolute temperature, *K* is the equilibrium constant, and *p* is the pressure [30].

Pressure can influence the equilibrium of a chemical reaction. Work is effectuated on the system by compression, which can be seen clearly in the diminishing of the intermolecular distances; thus, the internal energy of the system is raised. HHP activates the system in a different way than heat does, allowing for transformations that are unfavorable under atmospheric pressure. An example of this is the denaturization of enzymes by pressure, in which water molecules are imported to the internal “hydrophobic” cavities of proteins if the pressure is high enough. The intermolecular repulsions of the compressed non-polar residues that populate these cavities can be relieved by interstitial water molecules that offer an energy minimum in a conformation that is not favorable under normal pressure [32]. The changes in the intermolecular distances will also affect solvation energies and other enthalpic and entropic thermodynamic terms, so the system is likely to behave in a different way than what is predicted by using the atmospheric pressure thermodynamic values. The principle of the effect of pressure on the equilibrium of a reaction system is based on Le Ch atelier’s principle, which states, adapted to this case, that any phenomenon that entails a ∆*V* reduction (∆*V* < 0) is favored under high pressure and vice versa.

Reactions like ionization equilibria of weak electrolytes in aqueous media are promoted under high pressure as the formation of new charged species reorganizes the water around them to a more compact conformation where the dipoles are oriented towards the charged solutes. This causes an overall reduction of the volume of the system, called electroconstriction, and the reaction is promoted by increased pressure. The selection of an adequate buffer to carry out an enzymatic transformation under pressure must take this into account, as the increase in pressure may significantly alter the pH of buffers made with commonly used salts, such as borates or phosphates, with ranges of ∆*V* in the range of –24 to –32 mL/mol [11].

For reactions at equilibrium under pressure with ∆*V* values of −10 mL mol^−1^, like most covalent bond formation reactions, the change in equilibrium constant that occurs and increases product concentration can be depicted as in Figure 1 and Figure 2 [16].

### 2.2. Effects of HHP on Kinetics

Pressure can change the rate of enzyme-catalyzed reactions [12], with these being accelerated or decelerated [33], depending on the type of enzyme, substrate nature, temperature, and reaction time [34]. The interdependence of pressure and temperature on enzymes’ catalytic activity with an accelerated substrate conversion at increased temperatures may lead to new optimum reaction conditions concerning an improved overall reaction rate [35].

The Michaelis-Menten model of enzyme kinetics offers a framework for the description of enzyme performance based on the kinetic model parameters *K_M_* and *V_max_* [36]. The model is based on a simple one-substrate enzymatic transformation where enzyme *E* catalyzes the conversion of substrate *S* to product *P* through a two-step process consisting of the reversible formation of the enzyme-substrate complex *ES* in the binding step and the subsequent irreversible transformation and liberation of the product and the free enzyme, as displayed in Equation (4) [37].
(4)$$E+S\;\underset{k_{-1}}{\leftarrow }\overset{k_{1}}{\to }\;ES\;\overset{k_{2}}{\to }\;E+P$$

The mathematical treatment of this system, under the assumptions of steady state for ES and *k*_2_ << *k*_−1_, leads to Equation (5), which is a parametrical model of the system.
(5)$$v=\frac{V{\;}_{max}\left[S\right]}{K_{M}+\left[S\right]}=\frac{k{\;}_{2}\left[E_{T}\right]\left[S\right]}{K_{M}+\left[S\right]}$$

In this equation, *K_M_* is the Michaelis-Menten constant, *S* is the substrate concentration, *E_T_* is the total enzyme concentration (*E* + *ES*), *v* is the rate of reaction, and *k*_1_ is the forward rate constant for substrate binding, *k*_−1_ is the reverse rate constant for substrate binding, *k*_2_ (or *k_cat_*) is the catalytic rate constant, and the *ES* complex is also called the Michaelis complex [37].

The sensitivity of a chemical reaction rate to pressure depends on the absolute value of *Va* (activation volume) and is given by Equation (6), where *R* is the universal gas constant, *T* is the absolute temperature, *k* is the rate constant, and *p* is the pressure (MPa) [30].
(6)$${\left(\frac{\partial \ln k}{\partial p}\right)}_{T}=-\frac{Va}{RT}$$

The reaction rate is accelerated or decelerated by pressure when *Va* is negative (*Va* < 0) or positive (*Va* > 0), respectively [30]. A higher magnitude of *Va* (positive or negative) causes greater sensitivity of a chemical reaction to pressure, while reactions with *Va* = 0 are pressure-independent [30]. The pressure effect on the activity of an enzyme is determined by *Va* according to Equation (7) [21,38]:(7)$$In\;k=\ln\left(A\right)-\frac{p\;Va}{RT}$$
where *R* is the universal gas constant, *T* is the absolute temperature, *k* is the rate constant, *p* is the pressure (MPa), *Va* is the activation volume, and *A* is the pre-exponential factor. 

According to the transition state theory, *Va* is the difference between the partial molar volume of the transition state and the sum of the partial molar volumes of the reactants under the same conditions [39]. The interpretation of this value when applied to enzymatic systems is, however, not straightforward. Transition state theory was developed for reactions occurring in the gas state, which differs greatly from the chemical environment found in a high-pressure enzymatic system. Multiple steps, at least two in the simplest model, or the entire catalyzed reaction as depicted above, can be affected in the enzymatic process by an increase in pressure. Complex enzymatic reactions may have multiple transition steps with different *Va* values that together determine an overall apparent *Va* [11]. The exact meaning of this in the context of enzymatic systems is difficult to extract, but the relationship between pressure and rate constant can still give important phenomenological information within the set of established parameters of a complex system, especially when it comes to process intensification.

## 3. Effect of High Pressure on the Activity of Particular Enzymes

Enzyme-catalyzed reactions respond to pressure in different ways, making a case-by-case analysis necessary [9]. This section describes and summarizes (Appendix A—Appendix A) several examples of recent studies on the effect of HHP on enzymatic activity.

### 3.1. Alcohol Dehydrogenase (EC 1.1.1.1)

Alcohol dehydrogenases (ADHs) are zinc-containing enzymes that play an essential role in the conversion of alcohols to aldehydes or ketones [40], with possible applications in the production of chiral pharmaceuticals (Figure 3) [41].

The activity of ADH from *Thermoanaerobium brockii* was reported to increase at high pressure up to 100 MPa with a *Va* of −24 mL mol^−1^ for 2-pentanol and −36 mL mol^−1^ for cyclopentanol [23].

In a different study, the stereospecificity of ADH from *Thermoanaerobacter ethanolicus* was influenced by hydrostatic pressure (0.1–137.5 MPa) and different temperatures (20–52 °C), with 2-butanol, 2-pentanol, and 2-hexanol as substrates. In all cases, higher pressures favored the S enantiomer reaction over the R enantiomer, while a rise in temperature had the opposite effect. The difference was explained by the authors as being caused by the desolvation of one of the two pockets in the active site of the enzyme that leads to a favored conformation for one of the two substrates [42].

### 3.2. Formate Dehydrogenase (EC 1.2.2.1)

Formate dehydrogenases (FDH) catalyze the oxidation of the formate anion to carbon dioxide while simultaneously reducing NAD^+^ to NADH (Figure 4).

These enzymes belong to one of the most extensively studied protein families because they are essential to industry and several research fields. FDH is used in many industry sectors to catalyze numerous essential metabolic reactions that can be applied in CO_2_ fixation and nicotinamide recycling systems. In the research domain, they can be used as a model enzyme to study the general mechanisms of catalysis due to their reaction mechanism being very well known [43].

In a study conducted by Jaworek et al. (2021), high-pressure stopped-flow methodology was applied in conjunction with fast UV/Vis detection to investigate the kinetic parameters of FDH. In a neat buffer solution, the reaction rate increases by one order of magnitude by increasing the temperature from 25 to 45 °C and the pressure from ambient up to the MPa range (0.1–200 Mpa). The addition of particular co-solvents further doubled the reaction rate of the reaction, in particular the compatible osmolyte trimethylamine N-oxide (TMAO) and its mixtures with the macromolecular crowding agent dextran. The thermodynamic model Perturbed Chain Statistical Associating Fluid Theory was successfully applied within a simplified activity-based Michaelis-Menten framework to predict the effects of co-solvents on the kinetic efficiency of FDH by accounting for interactions involving substrate, co-solvent, water, and FDH. Especially mixtures of the co-solvents at high concentrations were beneficial for the kinetic efficiency and for the unfolding temperature [43].

### 3.3. Octopine Dehydrogenase (EC 1.5.1.11)

Octopine dehydrogenase (ODH) is found in scallops, and it is one of the very few monomeric NAD-dependent dehydrogenases that catalyze the condensation of pyruvate and L-arginine into octopine, as shown in Figure 5. This reaction provides the regeneration of NAD+ in anaerobic conditions, which contributes to the production of ATP during glycolysis [44].

This enzyme showed a complex pressure dependence on its catalytic activity, where the pressure dependence of the ODH activity was monitored for octopine synthesis and the oxidative degradation of octopine (reverse reaction), resulting in the formation of arginine and pyruvate. The maximum activity of ODH for octopine synthesis and degradation, monitored over a pressure range of 0.1–100 Mpa, was obtained at 60 Mpa Va values of −16.4 mL mol^−1^ and –11 mL mol^−1^, respectively [21]. 

### 3.4. Pectin Methylesterase (EC 3.1.1.11)

Pectin methylesterase (PME) catalyzes the demethylation of pectin, which leads to the formation of negative charges in this polysaccharide (Figure 6) [45].

This reaction is used in the industrial production of fruit and vegetable juices because it leads to a reduction in the viscosity of the juice and contributes to the clarification of the product by forming a low-methoxyl pectin precipitate [4]. Most studies regarding this enzyme are performed in the context of food science, so the diversity of chemical environments in each juice where the experiments take place may strongly influence the stability and activity of PME, as well as intrinsic differences between PMEs from different species.

The activity of PME-catalyzed pectin de-esterification was evaluated at different pressures (0.1–700 MPa) and temperatures (20–70 °C). Results showed that this reaction could be accelerated by combining elevated pressures with moderate heating to an optimum level of 200 or 300 MPa and 50–55 °C, with the enzyme showing to be always faster under pressure (maximum 700 MPa) than at atmospheric pressure (0.1 MPa) [46]. 

The reaction of pepper PME with pectin at different pressures (400–800 MPa) showed high temperature protection with increased pressure, with stable activity at 60 °C under 700 MPa. The same temperature at 0.1 MPa completely inactivates the enzyme [47]. On the other hand, maximum activity conditions for carrot PME in pressure ranges from 0.1–500 MPa and temperatures 20–65 °C were reported to be at 500 MPa and 50 °C with *Va* between −7.80 mL mol^−1^ and −5.72 mL mol^−1^ [48].

### 3.5. β-Glucanase (EC 3.2.1.2)

β-glucanase activity in malt plays an important role in brewing, affecting beer filtration [49]. During the barley malting process, partial hydrolysis of β-glucans by β-glucanase (Figure 7) initiates with seed germination, but the β-glucanases are heat inactivated at temperatures above 50 °C during the malting and mashing processes, whereas the remaining high molecular weight β-glucans may cause severe problems in beer processing and quality.

The reduction of β-glucans present in malt barley is highly desirable [51]. Buckow et al. (2005) concluded that high pressure (0.1–900 MPa) increases the thermostability of β-glucanase from barley malt at different temperatures (30–75 °C). A maximum of β-glucan depolymerization was observed at 55 °C, 215 MPa, and pH 5.6 after a 20 min reaction time. In this case, the depolymerization of β-glucan is approximately 1.6-fold higher than the maximum found at ambient pressure at 45 °C [49]. It is evident that HHP, in combination with elevated temperature, can be utilized to improve β-glucanase activity for depolymerizing β-glucans, opening new possibilities to accelerate the whole mashing process by the use of high pressure [49].

### 3.6. Cellulase (EC 3.2.1.4)

Cellulases are a group of enzymes that can convert lignocellulose, the most abundant and renewable source of energy on Earth [52], to glucose as the substrate for the production of many fermentation products, including a renewable alternative for fuel, bioethanol (Figure 8) [53].

Murao et al. (1992) [55] referred to increased activity of cellulases, namely Acucelase from *Aspergillus niger* and CM-cellulase, using as substrates microcrystalline cellulose (Avicel) and carboxymethyl cellulose (CM-cellulose), respectively, under high pressures ranging from 200 to 400 MPa at 37 °C. The activity of cellulase from *Aspergillus niger* obtained at 300 MPa was 1.5-fold higher than at 0.1 MPa, while CM-cellulase at 400 MPa increased 1.7-fold more than at 0.1 MPa. However, no *Va* values were mentioned to mathematically describe the effect of pressure on cellulase activity.

Similarly, Salvador et al. (2010) [38] showed that cellulase activity from *Aspergillus niger* at 30 °C increased from 60% at 200 MPa to 100% at 400 MPa (*Va* of −6.33 mL mol^−1^), compared to atmospheric pressure. In the second part of this work, the activity of cellulase was evaluated in the presence of the ionic liquid (IL) 1-butyl-3-methylimidazolium chloride ([bmim]Cl). At high pressure, the enzyme activity decreased with increasing concentration of [bmim]Cl, reaching 50% of the value in aqueous buffer with 20% [bmim]Cl. Nonetheless, in 10% [bmim]Cl under pressure, cellulase activity was improved compared to 0.1 MPa, varying from equal (at 600 MPa) to 1.7-fold higher (at 100 MPa). The *Va* obtained with 10% bmin[Cl] from 200 to 600 MPa was 1.58 mL mol^−1^, explaining that the reaction was slightly inhibited in this range of pressures, as seen in Figure 9. Nonetheless, the authors suggested that the results at 100 MPa could improve cellulase activity in ILs by carrying out the reaction under pressure. This was also the only study found in the literature concerning the activity of enzymes under pressure where the authors ensured that the possible effects of high pressure on the hydrolysis reaction were not due to pressure-induced changes in the substrate structure or pressure-induced enzyme inhibition. The authors could conclude this because they submitted the substrate and the enzyme separately to pressure and subsequently carried out the reaction at atmospheric pressure and reported no difference in the enzyme activity.

### 3.7. Naringinase (EC 3.2.1.40)

Naringinase has high potential in the food industry to sweeten fruit juice [56]. Naringin, a flavonoid responsible for bitterness in citrus fruit juices, is hydrolyzed by the α-L-rhamnosidase activity of naringinase, forming rhamnose and prunin, which can be further hydrolyzed by the β-D-glucosidase component of naringinase into glucose and the tasteless naringenin (Figure 10) [57].

Vila-Real et al. (2010) [36] tested the effects of high pressure (0.1–200 MPa) and different temperatures (25–80 °C) on α-L-rhamnosidase and β-d-glucosidase activities expressed by naringinase. Results showed a 3- and 4-fold increase in naringinase thermostability at 150 MPa and 70–80 °C, respectively, compared to 0.1 MPa and a 15-fold increase in kcat/K_M_ values from 0.1 MPa and 30 °C to 150 MPa and 70 °C. The *Va* values for α-L-rhamnopyranoside and naringinase reactions were −7.7 ± 1.5 and −20.0 ± 5.2 mL mol^−1^, respectively.

In a different study [58], the effects of high pressure (0.1–160 MPa) on the naringin bioconversion catalyzed by naringinase at 30 °C were evaluated. Hydrolyses of naringin were accelerated at 80 MPa (a 5% increase), and maximum activity was achieved at 160 MPa (a 20% increase) in comparison with 0.1 MPa with a *Va* of −15 mL mol^−1^.

Pedro et al. (2006) [59] studied the influence of high pressure (0.1–200 MPa) on the activity of an immobilized naringinase system at different temperatures (30–40 °C). A 2-fold activity increase with maximum activity at 160 MPa and at 35–40 °C was observed, with a *Va* of −9 mL mol^−1^.

In a study conducted by Marques et al. (2007) [60], response surface methodology was used to model the high pressure-temperature effects on naringin hydrolysis catalyzed by naringinase. Higher naringinase activity was obtained at higher pressure and temperature, with maximum activity being observed at 41 °C and 158 MPa.

### 3.8. Chymotrypsin (EC 3.4.21.1)

The enzyme α-chymotrypsin (CT) is a serine proteinase that enables the hydrolysis of peptide bonds, which is important to protein digestion (Figure 11) [4].

In a study conducted by Mozhaev et al. (1996), the effect of high pressure (0.1–500 MPa) and different temperatures (20–50 °C) on CT-catalyzed hydrolysis of N-succinyl-L-phenylalanine-*p*-nitroanilide was evaluated [7]. The acceleration effect of high pressure on CT-catalyzed hydrolysis of this substrate was more pronounced at high temperatures. For example, an increase in pressure (up to 470 MPa) at 20 °C raised the rate of enzymatic reaction 6.5-fold more than at 0.1 MPa, characterized by a negative *Va* value of −10 mL mol^−1^, whereas at 50 °C and 360 MPa, the activity of CT was 30-fold higher than at 20 °C and 0.1 MPa, with a higher negative *Va* of –25 mL mol^−1^ than at 20 °C. The influence of pressure and temperature and their combined effects on CT activity on the hydrolysis of N-succinyl-L-phenylalanine-*p*-nitroanilide can be found in the same article [7]. The reaction acceleration due to temperature increases is higher at higher pressures. The catalytic activity of CT at temperatures higher than 45 °C is significantly greater at elevated pressures than at 0.1 MPa. For instance, at 52.5 °C, the accelerating effect of a pressure of 180 MPa is nearly 20-fold [7].

Jaworek et al. (2018) studied the combined and separate effects of high pressure (0.1–200 MPa) and different co-solvents (glycine, TMAO, urea, dimethyl sulfoxide, deep sea buffer, and shrimp shallow buffer) at 20 °C using high pressure stopped-flow methodology in combination with fast UV/Vis detection on the activity of CT on N-succinyl-L-phenylalanine-*p*-nitroanilide. Results show an increase in CT activity with the increase in pressure until 200 MPa when the reaction occurred in pure Tris-HCl buffer, with the results being explained by the negative *Va* of −15 mL mol^−1^. When high pressure is combined with co-solvents, the effect on CT activity depends on the solvent used, but for most, high pressure also leads to an increase in enzyme activity until 200 MPa. These results might not only help to understand the modulation of enzymatic reactions by natural osmolytes but also elucidate ways to optimize enzymatic processes in biotechnological applications [62].

Schuabb et al. (2016) studied the effect of high pressure on the enzyme activity of immobilized CT on silica particles and free CT using high-pressure stopped-flow methodology. Reactions were carried out at different pressures (0.1–200 MPa), and product formation was increased at all pressures for immobilized and free CT. Additionally, product formation was higher for the immobilized CT (up to 8 times higher) when compared to the reactions performed under pressure but with free CT [63].

### 3.9. Trypsin (EC 3.4.21.4)

Trypsin is an important enzyme in protein digestion, catalyzing the degradation of long peptides into smaller ones (Figure 12) [64].

Trypsin is activated by moderate hydrostatic pressures, revealing a negative *Va* in the rate-limiting step of −8.8 mL mol^−1^ up to 43 MPa [21]. Similarly, a negative *Va* of −2.4 mL mol^−1^ in trypsin-catalyzed hydrolyses of N-benzoyl-L-arginine ethyl ester as substrate was observed in a pressure range of 0.1–100 MPa at 25 °C [65], with the results showing increased activity of trypsin with an increase in pressure up to 100 MPa.

Chicón et al. (2006) described the effect of pressure up to 400 MPa on the proteolysis of β-lactoglobulin A (β-Lg A) using trypsin. The hydrolysis of β-Lg A under pressure and β-Lg A previously treated with pressure was tested. Pressurization up to 400 MPa, before (only β-Lg A) or during enzyme action, enhanced tryptic hydrolysis of β-Lg A. Moreover, these authors concluded that β-Lg A was more susceptible to proteolysis when the enzymatic treatment was carried out under pressure [64].

### 3.10. Thermolysin (EC 3.4.24.27)

Thermolysin, one of the most studied enzymes, is a Zn-dependent protease [66] that, due to its high efficiency in the synthesis of peptides, mainly aspartame (Figure 13), an artificial peptide sweetener, has received growing interest in the biotechnological industry [22,67].

Pressure (0.1–300 MPa) and temperature (7–47 °C) dependence of the relative rate of thermolysin-catalyzed hydrolysis of the dipeptide amide 3-(2-furylacryloyl)-22-L-leucine amide (FA-Gly-Leu-NH_2_) and heptapeptide substrate MeOcAc-peptide showed acceleration at elevated pressures. At 300 MPa, for the dipeptide substrate, a 30-fold increase in the hydrolysation rate was observed when compared to 0.1 MPa, with the maximum rate observed at 220 MPa. When it comes to the heptapeptide substrate, only a 2-fold increase at 300 MPa was observed. However, at 150 MPa, its maximum rate was achieved, which was about 6-fold the value at 0.1 MPa. The K_M_ values for both substrates at 25 °C, pH 6.5, and atmospheric pressure were 4 µM, and the substrate concentrations were managed at less than 8% of the respective K_M_ values in the evaluations of the rates so it would not result in inhibition. Thermolysin-catalyzed hydrolysis of both substrates had a *Va* of 60 ± 10 mL mol^−1^. It was also possible to observe that an increase in temperature usually leads to a reduction of *Va*, which −70 mL mol^−1^ at 25 °C and 95 mL mol^−1^ at 45 °C [17].

Kudryashova et al. (1998) examined the catalytic activity of thermolysin from *Bacillus thermoproteolyticus* under pressure (0.1–400 MPa) and different temperatures (20–80 °C) by monitoring the hydrolysis of a 3-(2-furylacryloyl)-glycyl-L-leucine amide as substrate and concluded that the sensitivity of the catalytic reaction under high pressure becomes considerably higher at elevated temperatures. At 20 °C, an increase in pressure up to 100 MPa led to a significant 15-fold increase in the reaction rate, and the value of *Va* was –44 mL·mol^−1^. On the other hand, an increase in temperature to 40 °C and 60 °C caused the values of *Va* to be even more negative (−48 mL·mol^−1^ and −52.4 mL·mol^−1^, respectively); therefore, the accelerating effect of HHP increased [22].

## 4. Organocatalysis under Pressure

Besides enzymes, there are other types of catalysts whose activity can be enhanced by HPP. Organocatalysts are one of those cases, as there are already several different studies showing that organocatalyzed reactions under high pressure can achieve higher product yields with a lower reaction time when compared to the same reactions under atmospheric pressure [68,69,70]. Organocatalysis is a type of catalysis in which an organic compound named organocatalyst made up of carbon, hydrogen, nitrogen, oxygen, phosphorus, sulfur, and other nonmetal elements increases the speed of a chemical reaction. Organocatalysts are much smaller, cheaper, and more stable than enzymes, leading to reduced synthesis costs, compatibility with more functional groups, and an easier reaction design. They can also be used in a much wider range of physico-chemical conditions when compared to enzymes. Another critical feature of organocatalysis is the asymmetric synthesis that produces enantiomers in different proportions [19]. Enzymes are also capable of asymmetric synthesis but are limited to a few compounds. On the other hand, organocatalysts can be designed and constructed by mankind in a much simpler way when compared to enzymes, since they are simpler and smaller molecules. This allows the catalysis of a wider spectrum of reactions with high chemo, regio, stereo, and enantioselectivity. Some enantiomers can even be produced with purity rates around 95–99% [19]. This fact is so important that in 2021, Benjamin List and David Macmillan were awarded the Nobel Prize in Chemistry due to their work on asymmetric organocatalysis [20,70,71]

Kwiatkowski et al. (2011) managed to accelerate the asymmetric organocatalytic conjugation of nitromethane to β,β-disubstituted β-CF3 enones by performing the reaction under high pressure (800–1000 MPa). Many different γ-nitroketones were obtained after only 20 h with yields between 73–82% and high enantiomeric purity (96–99% ee). Similar results were obtained under atmospheric pressure, but only after 5 days of reaction time [69].

High pressure also managed to enhance an organocatalytic Friedel-Crafts alkylation of indoles with α, β-unsaturated ketones using primary amine salts as an organocatalyst. As pressure increased (100–1000 MPa), higher reaction yields were also obtained. The most promising result was the formation of the compound hexane–i-PrOH with a reaction yield of 86% and an enantiomeric purity of 90% after only 20 h. These are very promising results since this type of catalytic system is almost nonactive at atmospheric pressure [72].

Cholewiak et al. (2018) studied the effect of high pressure on the highly enantioselective 1,4 addition of nitromethane and 2-nitropropane to chalcones. Reactions were carried out at room temperature with pressures up to 900 MPa, which allowed to obtain a wide range of γ-nitroketones in only 1–5 h with yields and enantioselectivity up to 98%, whereas in control conditions (atmospheric pressure) only a very small amount of product (<10%) was obtained after 20 h [68].

For the asymmetric desymmetrization of 4,4-disubstituted cyclohexadienones, a highly diastereoselective and enantioselective approach was established using the Michael addition reaction of malonates catalyzed by the primary amine-thiourea conjugate catalyst and 4-pyrrolidino-pyridine under high pressure. The reaction was effectively accelerated under pressure (400–800 MPa) and products were obtained after only 2 days with yields up to 99% and enantioselectivity up to 93%, while under atmospheric pressure the maximum reaction yield obtained was 11% after 4.5 days [73].

Kasztelan and Kwiatkowski (2016) tested the effect of high pressure on the enantioselective hydroxyalkylation of indoles and 7-azaindole with trifluoromethyl ketones using cinchonidine as an organocatalyst. Reactions were carried out with pressures up to 900 MPa, and alcohols were obtained with high yields (97%) and high enantiomeric purity (80.5%), differently from the reactions carried out under atmospheric pressure, which only resulted in traces of product. The same reaction, but catalyzed using chiral phosphoric acid, was also pressure enhanced (900 MPa), obtaining alcohols after only 20 h with yields up to 93% and enantioselectivity up to 98%, contrary to what was observed with the reaction at atmospheric pressure, where after 36 to 96 h only a small amount of product was obtained (1–20%) [74].

## 5. Life’s Response to High Hydrostatic Pressure Environments

At the bottom of the ocean, about 11 km underwater in the deep Mariana Trench, pressure can be as high as 100 MPa. Complex ecosystems are found there, which, like any other known life form, rely on the catalytic action of enzymes. Enzymes from this evolutionary context have gotten the attention of studies oriented towards the understanding of proteins’ adaptations to extreme pressure. In an exploration of the adaptability of microbial life to high pressure, Abe (2007) set hydrostatic pressure limits for life-sustaining processes such as cell division (20 MPa), protein synthesis (50 MPa), and soluble enzyme function (100 MPa) [75]. However, some organisms have been found to survive at higher pressures; the current record holder is *Thermococcus piezophilus*, which is able to survive up to 125 MPa [76], defying the previous pressure limits of life. 

Gerringer et al. (2020) [77] studied lactase dehydrogenase (LDH) and malate dehydrogenase (MDH) in muscle tissue of several fish species living at different depths. The LDH activity of the deep living fish studied increased with pressure, while the enzymes of the shallow species decreased in activity. This was interpreted by the authors as a suggestion of “evolutionary changes in LDH ∆*V*″ in the proteins of fish that have adapted to higher depths. In contrast, MDH showed higher activity levels with increasing pressure for all the studied species, which in turn suggests an intrinsically pressure-promoted mechanism. Siebenaller and Somero’s work [78] showed a similar behavior for other species: a markedly steeper increase in K_M_ for pyruvate and NADH with increasing pressure in the case of shallow species, while enzymes from the deep-sea species were able to better retain the affinity for their substrates. These enzymes have almost identical primary structures, and only 21 amino acids were found to be different between species. The main difference between these enzymes was found in the N-terminus of the protein, which is probably the main contributor to the pressure resistance of the enzyme isoforms from the deep-sea species. Gerringer (2020) reported that the adaptation of proteins to high pressure in marine animals is partial and that their resistance is not necessarily intrinsic to the proteins themselves but that there are several adaptation mechanisms working simultaneously. Crustaceans and fish accumulate small organic molecules that serve as protecting agents against the increasing pressure, denominated piezolytes. Among them, the main example is TMAO, which shows a linear increase in concentration levels in the tissues with regards to increasing depth [77]. Gerringer et al. (2020) showed an example of the counteraction of pressure-induced inhibition in the pyruvate kinase of a hadal fish species, *Notoliparis kermadecensis*, by TMAO at a concentration of 400 mM [77].

Bacterial lines have been shown to possess the capability of TMAO synthesis. The work of Qin et al. (2021) recently proved the existence of a trimethylamine transporter protein and a monooxygenase that allow the intracellular accumulation of TMAO [79]. Heterologous expression of this system in *Escherichia coli* and *Bacillus subtilis* was successful and improved survival and growth under high pressure (20 and 40 MPa), as observed in both cases. Another observed adaptation mechanism is the increased expression of enzymes in the muscle tissue of deep-living species due to compromised catalytic activity at higher pressures [77].

Overall, nature puts in place multiple extrinsic and intrinsic adaptation mechanisms for the correct function of enzymes at high pressure. There is no unique molecular signal for HHP adaptation, and not all of the possible mechanisms have been thoroughly explored [80]. Further research in this area may open up new possibilities for bioindustrial processes with pressure-resistant microorganisms. The enzymatic studies of extremophile metabolisms also offer high value to astrobiology, as they help to better define the line between hospitable and unhospitable environments and may expand the places considered for potential life outside of this planet.

## 6. Conclusions

From this review, it is noteworthy that HHP can be a feasible tool to manipulate and modulate enzymatic reactions. Indeed, the enzyme-catalyzed reaction activity can accelerate or decelerate in combination with moderate/elevated temperatures under HHP conditions. The studies here highlighted and discussed can improve the enzymatic activity of the reaction and be a starting point for new possibilities in the food and other industries, such as accelerating enzyme-mediated processes in the food industry, which could lead to a significant improvement in the product’s quality. The yield of product concentration at equilibrium under pressure can be significantly improved, which is a great advantage in economic terms as more products from the enzymatic reaction are produced without increasing the amount of substrate. Thus far, the experimental conditions under which the enzymatic reactions take place need to be carefully controlled, as it is expected that some pH shifts will occur on the buffers under HHP. Even though the current literature points to a couple of examples of how much the pH of a couple of buffers shifts under pressure, this is not always considered during the design of the experiments.

However, more research is to be conducted on an enzyme-by-enzyme basis since the effect of HHP, alone or in combination with temperature, can be quite different for a particular enzyme, and the effects of HHP on the structure of the enzyme are to be accessed to understand, from a fundamental point of view, how HHP-mediated enzymatic reactions can occur in extreme environments, such as those found in the deep sea. Other catalyzed reactions (such as organocatalysis) mediated by HHP are also very pertinent topics of research, with the potential of HHP remaining unexplored in this field.

## Figures and Tables

**Figure 1 molecules-28-04172-f001:**
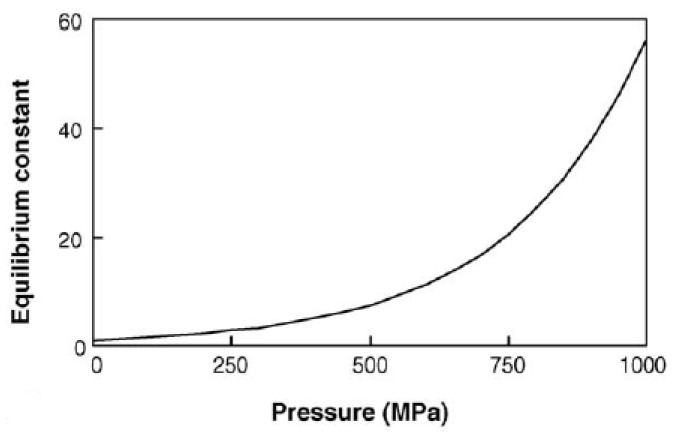
Improvement of the equilibrium constant at different pressures: *K*_0.1MPa_ = 1; *T*= 25 °C; ∆*V*= −10 mL mol^−1^. Adapted from [16].

**Figure 2 molecules-28-04172-f002:**
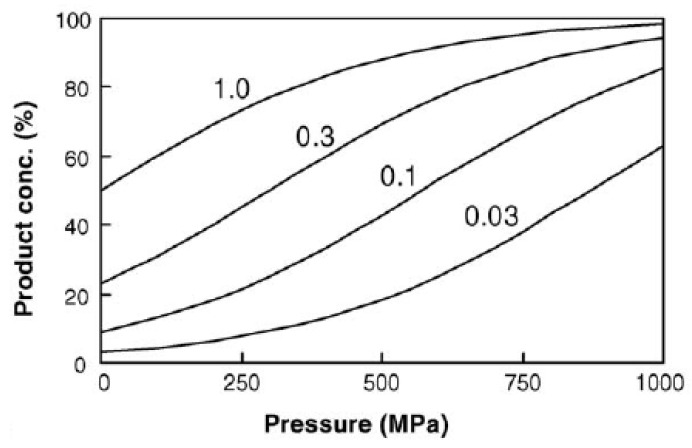
Increase in product concentration at equilibrium as a function of pressure for a condensation of A + B→C; T = 25 °C; ∆*V* = −10 mL mol^−1^; from top to bottom line: *K*_0.1MPa_ = 1.0, 0.3, 0.1, and 0.03. Adapted from [16].

**Figure 3 molecules-28-04172-f003:**
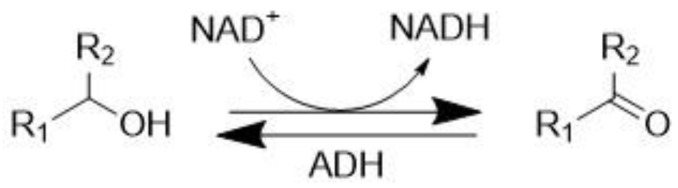
Alcohol dehydrogenase (ADH) catalyzed conversion of alcohols to carbonyls.

**Figure 4 molecules-28-04172-f004:**
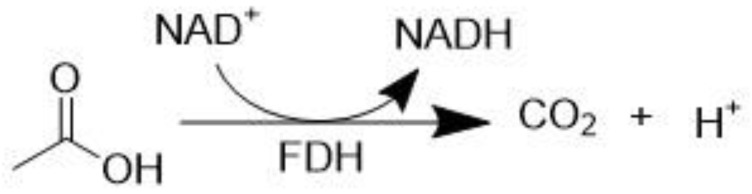
Formate dehydrogenase (FDH) catalyzed oxidation of formate to carbon dioxide.

**Figure 5 molecules-28-04172-f005:**
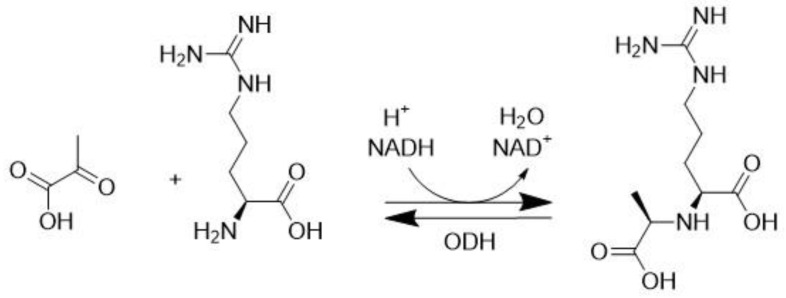
Octopine dehydrogenase (ODH) catalyzed synthesis of octopine.

**Figure 6 molecules-28-04172-f006:**
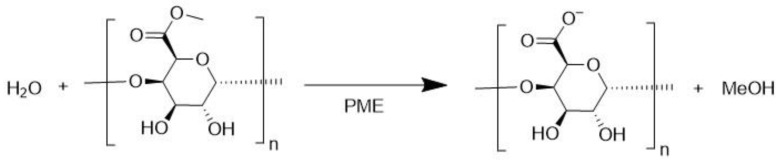
Pectin methylesterase (PME) catalyzed demethylation of pectin with methanol formation.

**Figure 7 molecules-28-04172-f007:**
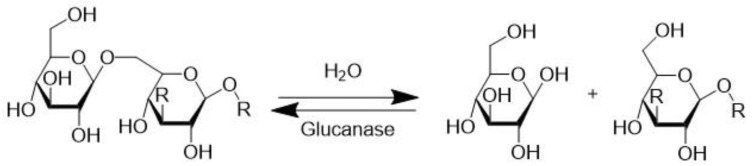
β-glucanase catalyzed hydrolysis/condensation of β-glucans, acting on a 1–6 bond [50].

**Figure 8 molecules-28-04172-f008:**
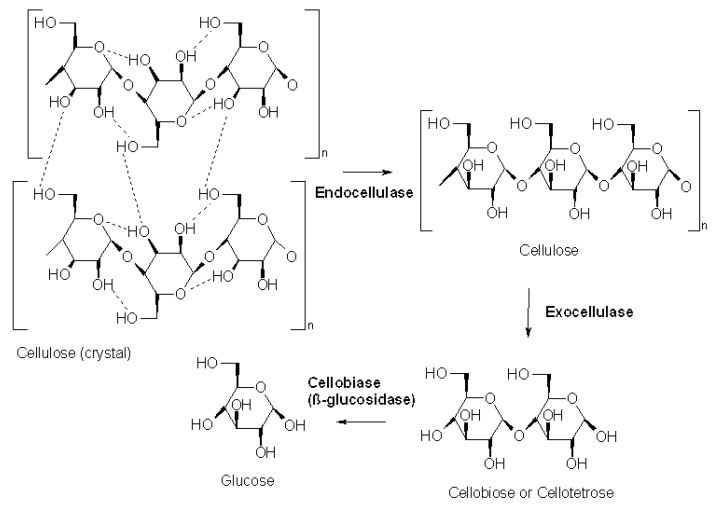
Cellulase catalyzed conversion of cellulose into glucose [54].

**Figure 9 molecules-28-04172-f009:**
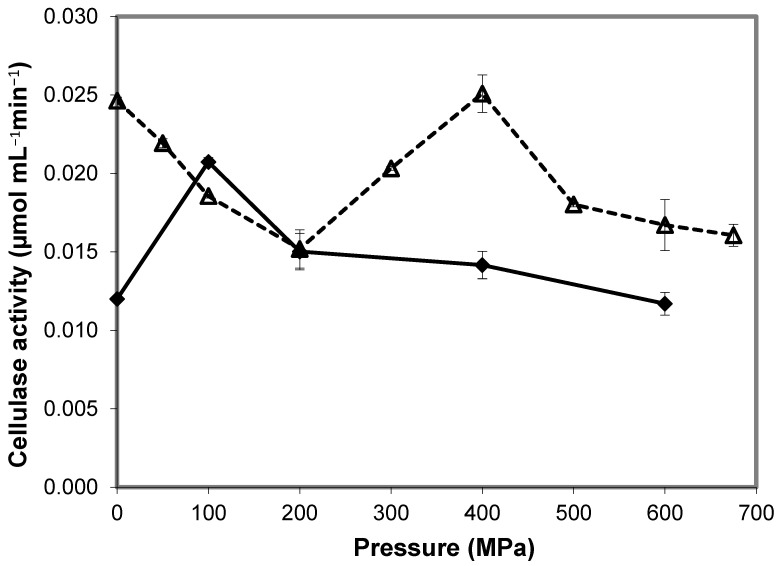
Effect of high pressure on cellulase activity in buffer (△) and in 10% [bmim]Cl (✦). Adapted from [38].

**Figure 10 molecules-28-04172-f010:**
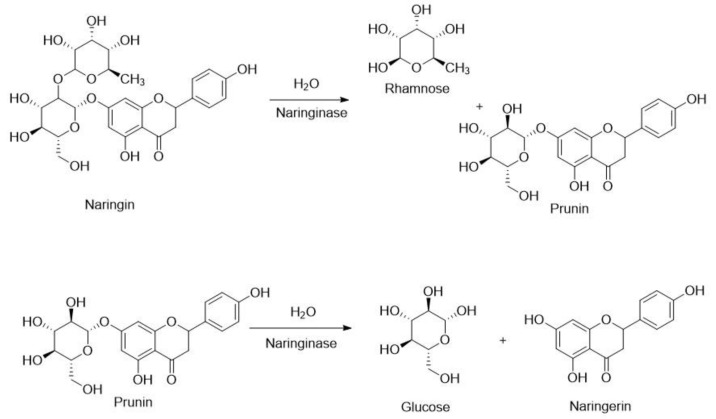
Two-step naringinase catalyzed conversion of naringin into the tasteless naringenin.

**Figure 11 molecules-28-04172-f011:**
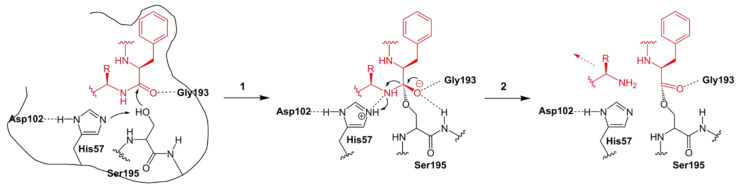
Mechanism of peptide hydrolysis catalyzed by α-chymotrypsin [61].

**Figure 12 molecules-28-04172-f012:**
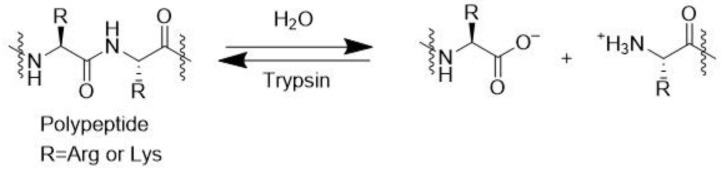
Peptide hydrolysis catalyzed by the enzyme trypsin.

**Figure 13 molecules-28-04172-f013:**
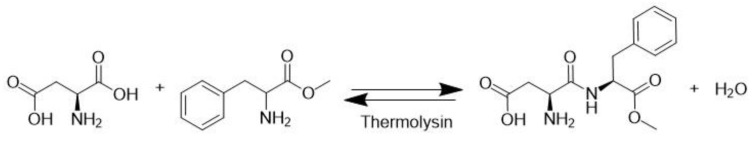
Aspartame synthesis catalyzed by the enzyme thermolysin.

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
