# Peer review of "High Hydrostatic Pressure in the Modulation of Enzymatic and Organocatalysis and Life under Pressure: A Review"

_molecules, 2023, doi:10.3390/molecules28104172_

Round 1

Reviewer 1 Report

The article provides a complete review of the effect of high hydrostatic pressure on the activity of various enzymes of interest due to their involvement in different metabolic and deterioration processes, as well as their possible applications in biotechnology. The effect of pressure on organocatalyst processes is also described. The article is well written and structured and could be considered suitable for publication in its current form. However, it is suggested to use the same type of pressure units throughout the document (preferred MPa versus kbar), and to put the full name of the microorganisms the first time they are cited (eg. Ln 506).

Author Response

Author’s answers to reviewers’ comments to the manuscript: molecules-2291359, entitled: High Hydrostatic Pressure in the Modulation of Enzymatic Activity: A Review

Point by point, author’s answers to the reviewers’ comments are shown just after each comment of the reviewers in green background text.

The changes made in the revised manuscript considering the reviewers comments are indicated by red color text.

Reviewer #1

The article provides a complete review of the effect of high hydrostatic pressure on the activity of various enzymes of interest due to their involvement in different metabolic and deterioration processes, as well as their possible applications in biotechnology. The effect of pressure on organocatalyst processes is also described. The article is well written and structured and could be considered suitable for publication in its current form. However, it is suggested to use the same type of pressure units throughout the document (preferred MPa versus kbar), and to put the full name of the microorganisms the first time they are cited (eg. Ln 506).

Changed in the whole manuscript as suggested.

Reviewer 2 Report

The manuscript make a reviewing of literature very complete, with well handly of language. However, literature is not very current. 

We think that title must be widened, because some sections are not included in this. 

The section of thermodynamic aspects, must be more deeply treated.

Some commets are on pdf field.

Author Response

Author’s answers to reviewers’ comments to the manuscript: molecules-2291359, entitled: High Hydrostatic Pressure in the Modulation of Enzymatic Activity: A Review

Point by point, author’s answers to the reviewers’ comments are shown just after each comment of the reviewers in green background text.

The changes made in the revised manuscript considering the reviewers comments are indicated by red color text.

Reviewer #2

The manuscript make a reviewing of literature very complete, with well handly of language. However, literature is not very current.

Concerning the effect of pressure on the reaction rate constant and equilibrium constant (this meaning studies that carry out the catalytic reaction under pressure), the authors could not find any other references; in literature one can see references but related to the effect of pressure on the enzyme to study enzyme inactivation.

We think that title must be widened, because some sections are not included in this.

Thank you for your comments.

The title was changed and now reads:

“High Hydrostatic Pressure in the Modulation of Enzymatic and Organocatalysis and Life Under Pressure: A Review”

The section of thermodynamic aspects, must be more deeply treated.

The thermodynamics section was expanded by adding examples to illustrate more clearly the points discussed.

Some commets are on pdf field.

Comments in the PDF were addressed in the PDF for the sake of easier visualization (with some parts of the text in the manuscript being also changed) and so, please, we ask the referee to see the comments in the PDF.

Reviewer 3 Report

Reviewer comments:

In the manuscript (molecules-2291359) the authors gave the overview how high hydrostatic pressure can also positively affect the catalytic action of some enzymes which are important for the industrial biotechnology. It is known that some enzymes can be inactivated during high pressure, which is very important from the aspect of food industry where HHP technology was applied as a novel non-thermal method of food pasteurization. But discovering that HHP can have a beneficial effect on the activity of some enzymes is a new area of scientific research. This beneficial effect can be on the level of reaction's equilibrium or the rate of reaction catalyzed by enzyme. Beside enzymes, the authors also included the effect of HHP on the organocatalysts and the reaction's equilibrium and rate of the reactions catalyzed by them. These artificial catalysts, constructed by humans, are more versatile than enzymes and can be promising catalysts which will have a significant role in the future bioindustry. The last part of review is dedicated to the studies which are deal with organisms and their adaptations to HHP environments, especially from the aspect of enzyme adaptation and their capability to synthetize small molecules like TMAO which are essential for the enzyme activity under HHP.

This article can be very interesting for the readers because beside already mention enzymes in this manuscript this field of research can be applied for many other enzymes.  

In this form, the manuscript needs minor corrections before the final decision. Please, take into account below some of the comments and suggestions for the improvement of your manuscript quality. In the pdf file of the manuscript, the text for the correction has been marked.

The main text

Lines 50-51

Please, correct the sentence according to the cited references.

Line 71

Please, insert at least one of their references (List and MacMillan).

Line 78

Please, replace high hydrostatic pressure with HHP.

Line 81

Replace pressure with HHP.

Line 82

Please, replace a reaction with an enzyme-catalysed reaction.

Line 83

Please replace the sentence A second section will give the reader an up-to-date overview of high pressure enzymatic studies

A second section will give the reader an up-to-date overview of the effect of HHP on the activity of particulate enzymes.

Lines 88-89

Suggestion: The protein structure, state of solvent media, thermodynamics, and kinetics of enzymatic systems can be affected by HHP.

Line 115, 119

Please correct in the manuscript ml into mL

Lines 119 and 123

at 0.1 MPa put as subscript.

Line 132

Insert of enzyme kinetics after model.

Line 192

Please, insert of enzyme  after active site

Figures 3,4 5, 

Please, correct the equation for the chemical reactions where they are reversible. Add H+ in Figure 5. Cofactors NAD+ and NADH in figure 4 should be put above the arrow.

Lines 211 and 212

Delete PC-SAFT

Line 216

Add of FDH after efficiency

Line 221

Please, insert into octopine after L-arginine.

Line 223

Please, replace sentence This provides the organism a means of sustaining  ATP production with an anaerobic source of NAD+. with 

This reaction provides the regeneration of NAD+ in anaerobic conditions which contributes to production ATP during glycolysis.

Lines 236

This reduces viscosity and forms low-methoxyl pectin precipitates causing clarification of cloudy fruit and vegetable juices.

Replace with

This reaction is used in the industrial production of fruit and vegetables juices, because it leads to reduction of viscosity of juice and contributes to the clarification of product by forming low-methoxyl pectin precipitate.

Line 252

Please, remove the point and space before glucanase.

Figure 10

The sugar part of naringin is the second product. Please, correct that at Figure 10, or prepare the new one in which both steps of naringin hydrolysis are shown.

Lines 333 and 336

Replace C with c in chymotrypsin.

Line 336

Insert the mechanism of  before peptide.

Line 352

Replace G with g in glycine. DMSO full name.

Lines 362,435,447, 504

Add the point after al.

Line 370

Replace long chains of amino acids into smaller chains with … long peptides into smaller ones.

Line 375

Please, delete (BzArgOEt)

Lines 379-383

Please replace β-Ig A with β-Lg A

Line 386

Delete precursor.

Line 402, 413

Delete marked part of the text.

Line 408

Replace with FA-Gly-Leu-NH2.

Line 414

Insert the reference number after increased.

Line 434

Please, cite some references of these authors.

Line 445

Replace this with these.

Line 491-493

These enzymes were not fundamentally different, with only 21 amino acid differences, being the region of the N-terminus of the protein the main contributor to pressure resistance.

Replace with

These enzymes have almost identical primary structure, only 21 amino acids were found to be different between species. The main difference between these enzymes was found in the N-terminus of the protein which probably is the main contributor to the pressure resistance of enzyme`s isoforms from the deep-sea species.

Line 494

Replace Yancey with Gerringer.

After proteins add to the high pressure.

References

The reference 37 should be formatted like other references.

Author Response

Author’s answers to reviewers’ comments to the manuscript: molecules-2291359, entitled: High Hydrostatic Pressure in the Modulation of Enzymatic Activity: A Review

Point by point, author’s answers to the reviewers’ comments are shown just after each comment of the reviewers in green background text.

The changes made in the revised manuscript considering the reviewers comments are indicated by red color text.

Reviewer #3

In the manuscript (molecules-2291359) the authors gave the overview how high hydrostatic pressure can also positively affect the catalytic action of some enzymes which are important for the industrial biotechnology. It is known that some enzymes can be inactivated during high pressure, which is very important from the aspect of food industry where HHP technology was applied as a novel non-thermal method of food pasteurization. But discovering that HHP can have a beneficial effect on the activity of some enzymes is a new area of scientific research. This beneficial effect can be on the level of reaction's equilibrium or the rate of reaction catalyzed by enzyme. Beside enzymes, the authors also included the effect of HHP on the organocatalysts and the reaction's equilibrium and rate of the reactions catalyzed by them. These artificial catalysts, constructed by humans, are more versatile than enzymes and can be promising catalysts which will have a significant role in the future bioindustry. The last part of review is dedicated to the studies which are deal with organisms and their adaptations to HHP environments, especially from the aspect of enzyme adaptation and their capability to synthetize small molecules like TMAO which are essential for the enzyme activity under HHP.

This article can be very interesting for the readers because beside already mention enzymes in this manuscript this field of research can be applied for many other enzymes. 

In this form, the manuscript needs minor corrections before the final decision. Please, take into account below some of the comments and suggestions for the improvement of your manuscript quality. In the pdf file of the manuscript, the text for the correction has been marked.

The main text

Lines 50-51

Please, correct the sentence according to the cited references.

Sentence corrected according to the cited reference [11] which used to read "later, the research evolved with Morild (1981) [11] who tabulated positive and negative effects on apparent activity for 135 enzymes, even though the results were not conclusive” and now reads “later, the research evolved with Morild (1981) [11] who made an extensive review of the theoretical knowledge, known until the present date, of the pressure effects on enzymes”.

Line 71

Please, insert at least one of their references (List and MacMillan).

New reference added of Benjamin List [20].

Line 78

Please, replace high hydrostatic pressure with HHP.

Changed accordingly.

Line 81

Replace pressure with HHP.

Changed accordingly.

Line 82

Please, replace a reaction with an enzyme-catalysed reaction.

Changed accordingly.

Line 83

Please replace the sentence A second section will give the reader an up-to-date overview of high pressure enzymatic studies

A second section will give the reader an up-to-date overview of the effect of HHP on the activity of particulate enzymes.

Changed accordingly with some alterations. It now reads “A second section will give the reader an up-to-date overview of the effect of HHP on the activity of particular enzymes”. Instead of “particulate” it now reads “particular”.

Lines 88-89

Suggestion: The protein structure, state of solvent media, thermodynamics, and kinetics of enzymatic systems can be affected by HHP.

Changed according to the suggestion.

Line 115, 119

Please correct in the manuscript ml into mL

Changed accordingly in the whole manuscript.

Lines 119 and 123

at 0.1 MPa put as subscript.

Changed accordingly.

Line 132

Insert of enzyme kinetics after model.

Changed accordingly.

Line 192

Please, insert of enzyme  after active site

Changed accordingly.

Figures 3,4 5,

E

Please, correct the equation for the chemical reactions where they are reversible. Add H+ in Figure 5. Cofactors NAD+ and NADH in figure 4 should be put above the arrow.

Changed accordingly.

Lines 211 and 212

Delete PC-SAFT

Changed accordingly.

Line 216

Add of FDH after efficiency

Changed accordingly.

Line 221

Please, insert into octopine after L-arginine.

Changed accordingly.

Line 223

Please, replace sentence This provides the organism a means of sustaining  ATP production with an anaerobic source of NAD+. with

This reaction provides the regeneration of NAD+ in anaerobic conditions which contributes to production ATP during glycolysis.

Changed accordingly.

Lines 236

This reduces viscosity and forms low-methoxyl pectin precipitates causing clarification of cloudy fruit and vegetable juices.

Replace with

This reaction is used in the industrial production of fruit and vegetables juices, because it leads to reduction of viscosity of juice and contributes to the clarification of product by forming low-methoxyl pectin precipitate.

Changed accordingly.

Line 252

Please, remove the point and space before glucanase.

Changed accordingly.

Figure 10

The sugar part of naringin is the second product. Please, correct that at Figure 10, or prepare the new one in which both steps of naringin hydrolysis are shown.

Lines 333 and 336

Replace C with c in chymotrypsin.

Changed accordingly.

Line 336

Insert the mechanism of  before peptide.

Changed accordingly.

Line 352

Replace G with g in glycine. DMSO full name.

Changed accordingly.

Lines 362,435,447, 504

Add the point after al.

Changed accordingly.

Line 370

Replace long chains of amino acids into smaller chains with … long peptides into smaller ones.

Changed accordingly.

Line 375

Please, delete (BzArgOEt)

Changed accordingly.

Lines 379-383

Please replace β-Ig A with β-Lg A

Changed accordingly.

Line 386

Delete precursor.

Changed accordingly.

Line 402, 413

Delete marked part of the text.

Line 402, the marked part was deleted and in line 413 authors changed the text to better explain the idea:

“On the other hand, an increase in temperature to 40 °C and 60 °C caused the values of Va to be even more negative (-48 mL.mol-1 and -52,4 mL.mol-1, respectively), therefore, the accelerating effect of HHP increased [22]”.

Line 408

Replace with FA-Gly-Leu-NH2.

Changed accordingly.

Line 414

Insert the reference number after increased.

Changed accordingly.

Line 434

Please, cite some references of these authors.

References [20] and [71] added.

Line 445

Replace this with these.

Changed accordingly.

Line 491-493

These enzymes were not fundamentally different, with only 21 amino acid differences, being the region of the N-terminus of the protein the main contributor to pressure resistance.

Replace with

These enzymes have almost identical primary structure, only 21 amino acids were found to be different between species. The main difference between these enzymes was found in the N-terminus of the protein which probably is the main contributor to the pressure resistance of enzyme`s isoforms from the deep-sea species.

Changed accordingly.

Line 494

Replace Yancey with Gerringer.

After proteins add to the high pressure.

Changed accordingly.

References

The reference 37 should be formatted like other references.

Changed to the same format.
